# Chikungunya virus in Europe: A retrospective epidemiology study from 2007 to 2023

Qian Liu[1], Hong Shen[2], Li Gu[2], Hui Yuan[1], Wentao Zhu[2]*

1 Department of Clinical Laboratory, Beijing Anzhen Hospital, Capital Medical University, Beijing, PR China, 2 Department of Infectious Diseases and Clinical Microbiology, Beijing Institute of Respiratory Medicine and Beijing Chao-Yang Hospital, Capital Medical University, Beijing, PR China

* wentaozhu@126.com

## Abstract

### Background

Chikungunya virus (CHIKV), a mosquito-borne alphavirus, is responsible for disease outbreaks worldwide. However, systematic knowledge of spatiotemporal distribution and risk patterns of CHIKV in mainland Europe remains unclear. Our aim was to decipher the epidemiological characteristics, diversity, and clinical manifestations of CHIKV.

### Methods

In this retrospective study, we retrieved the surveillance bulletins of chikungunya infections reported in Europe during 2007–2023 to depict the epidemiological characteristics. We performed genotyping and phylogenetic analyses to examine the evolution and mutation. We also searched PubMed, Web of Science, and Google Scholar to conduct meta-analyses of clinical manifestations.

### Results

4730 chikungunya cases across twenty-two countries were documented in mainland Europe from 2007–2022, with no cases reported in 2023. The age-standardized incidence rate was highest in 2014 (0.31), with significant variations observed in each country per year. Although autochthonous outbreaks occurred in several countries, the majority of cases were travel-related, with individuals mainly getting infected during summer vacation. Most travel-related cases were reported as being acquired in India (11.7%), followed by Dominican Republic (9.0%), Guadeloupe (8.7%), and Thailand (7.8%). Genotyping of genome sequences identified two genotypes, with the majority belonging to II-ECSA. The E1 A226V mutation was detected from autochthonous outbreaks, including Italy in 2007 and France in 2014 and 2017. The most common symptoms reported were fever (97.6%), joint pain (94.3%), fatigue (63.5%), and skin rash (52.3%).

### Conclusion

The suitable niches for CHIKV are expanding due to climate change and global travel. With the absence of specific antiviral treatments and vaccines still in development, surveillance and vector control are essential in suppressing the re-emergence and epidemics of CHIKV.

**Data availability statement:** The sequences for the study are retrieved from the publicly

available GISAID database, with virus names and accession numbers provided in the supplementary excel spreadsheet. The accession numbers include EPI_ISL_17454873 to EPI_ISL_17454876, EPI_ISL_17455499, EPI_ISL_17455726, EPI_ISL_17455796, EPI_ISL_17455798 to EPI_ISL_17455800, EPI_ISL_17455803, EPI_ISL_17455804, EPI_ISL_17455808, EPI_ISL_17455809, EPI_ISL_17455812 to EPI_ISL_17455819, EPI_ISL_17455821 to EPI_ISL_17455825, EPI_ISL_17455827 to EPI_ISL_17455837, EPI_ISL_17455839_EPI to ISL_17455844, EPI_ISL_17458068, EPI_ISL_17458069, EPI_ISL_17458071 to EPI_ISL_17458073, EPI_ISL_17458082, EPI_ISL_17458415, EPI_ISL_17458684, EPI_ISL_17458751, EPI_ISL_17458755, EPI_ISL_17458920, EPI_ISL_17459935, EPI_ISL_17459939 to EPI_ISL_17459941, EPI_ISL_17459944 to EPI_ISL_17459946, EPI_ISL_17459948, EPI_ISL_17459949, EPI_ISL_17459951 to EPI_ISL_17459955, EPI_ISL_17459957, EPI_ISL_17460710, EPI_ISL_17461258, EPI_ISL_17461259, EPI_ISL_17461283 to EPI_ISL_17461286, EPI_ISL_17461337 to EPI_ISL_17461340, EPI_ISL_17461546 to EPI_ISL_17461546. All epidemiological data used in this study was retrieved from the European Center of Disease Prevention and Control (https://www.ecdc. europa.eu/en/publications-data/monitoring/ all-annual-epidemiological-reports).

**Funding:** This work was supported by the Beijing Chao-Yang Hospital Golden Seeds Foundation (CYJZ202220 to W.-T. Z.). The funders had no role in study design, data collection and analysis, decision to publish, or preparation of the manuscript.

**Competing interests:** The authors have declared that no competing interests exist.

## Author summary

This study provides the first overview of chikungunya infections in mainland Europe in the time period 2007–2023. The 4730 chikungunya cases were reported in 22 countries, predominantly affecting females aged 45–64 years. The United Kingdom had the highest number of cases (21.9%), followed by France (19.6%), Germany (14.5%), and Italy (13.6%). Bubble plot illustrating the dynamics of age-standardized incidence rates by country per year showed significant variation. The majority of chikungunya infections in humans were travel-related, with most cases originating from India (11.7%), Dominican Republic (9.0%), Guadeloupe (8.7%), and Thailand (7.8%). Chikungunya cases in Europe showed a seasonal pattern with peaks mainly associated with summer vacation. Two genotypes were identified based on genome sequencing, with the II-ECSA genotype being predominant. Common symptoms included fever (97.6%), joint pain (94.3%), and fatigue (63.5%).

## Introduction

The chikungunya virus (CHIKV), a member of the genus *Alphavirus* within the family *Togaviridae*, is an enveloped and positive-strand arbovirus [1]. CHIKV is primarily transmitted to humans through the bite of infected female mosquitoes, most commonly *Ae. albopictus* and *Ae. Aegypti* [2]. Currently, CHIKV is classified into three major genotypes: West African, East-Central-South-African (ECSA), and Asian [3]. The Indian Ocean lineage (IOL), emerging from the ECSA genotype, has been responsible for epidemics since 2005 in the Indian Ocean islands, as well as south and southeast Asia, and Europe [1,4]. The adaptive mutations in the E1 and E2 virus envelope glycoproteins of ECSA-IOL have facilitated adaptation for *Ae albopictus* infection and transmission [5].

Chikungunya was first recognized during an outbreak in southern Tanzania in 1952 [6]. The largest epidemic ever recorded began on Lamu Island, Kenya, in 2004 [1]. By 2007, CHIKV was imported into mainland Europe, causing an outbreak of chikungunya fever in Italy [7]. This suggested for the first time the significant potential of the virus to move to novel ecological niches, including Europe, Australia, and the Western Hemisphere [8]. Over the past 20 years, more than 10 million cases of chikungunya have been reported in over 125 countries or territories in Asia, Africa, Europe and the Americas [1,2], with 1.3 billion people being at risk of chikungunya [9]. Although some outbreaks were reported in France and Italy, chikungunya is not endemic in the European Union (EU), as it is mainly introduced by travelers into receptive areas [7,10,11].

Most people infected with CHIKV will develop symptoms and enter an acute phase with symptoms such as sudden onset with high fever (> 39°C), joint pain, skin rash, myalgia, headache, and fatigue [12]. Deaths from chikungunya are rare. A significant proportion of CHIKV patients have reported long-term sequelae, with persistent arthralgia, arthritis, alopecia, and depression being the most commonly mentioned [13]. Currently, there are no licensed vaccines or antiviral drugs available to prevent CHIKV infection or treat chikungunya [14], leading to high public health costs [12]. The re-emergence and spread of CHIKV to new regions pose a growing global public health threat [15]. Despite this, awareness and understanding of chikungunya remain limited among affected populations and healthcare workers [12,16].

In this study, we conducted a comprehensive examination of the history and spatiotemporal distribution of CHIKV infection in Europe based on multiple analyses of surveillance data. Additionally, we explored the potential risks of chikungunya in Europe and summarized the clinical manifestations through a literature review.

## Methods

### Data collection

The surveillance data for CHIKV cases, including age, sex, age-standardized incidence rate, and reporting countries, were retrieved from online archives of Eurosurveillance from European Center of Disease Prevention and Control (ECDC) and published data.

A case must meet at least one of the following four criteria to be considered a confirmed case: CHIKV isolated from a clinical specimen, detection of virus nucleic acid, chikungunya specific IgM antibodies were detected and confirmed by neutralization, seroconversion or four-fold antibody titer increase of chikungunya specific antibodies [17]. Probable cases were defined as the presence of chikungunya specific IgM antibodies in a single serum sample [17]. Travel-related cases were defined as those in which individuals in European countries (reporting countries) had a probable or confirmed chikungunya infection acquired outside their country of residence [10].

### Genotyping and phylogenetic analysis

We retrieved all genome records from the Global Initiative on Sharing Avian Influenza Data (GISAID, https://gisaid.org/) that met the criteria of being location in Europe, being of host human, and having a collection date before 2023. The genotypes were determined using the Genome Detective Chikungunya Typing Tool [18]. To conduct a phylogenetic analysis of CHIKV, the genomes were aligned using MAFFT v. 7.526 with FFT-NS-2 algorithm (https://mafft.cbrc.jp/alignment/software/). After trimming the 5' and 3' ends, the GTR nucleotide substitution model was predicted as the best-fit model using the ModelFinder [19]. A maximum likelihood (ML) phylogenetic tree was inferred using IQ-TREE2 [20] and further annotated using Interaction Tree of Life (iTOL, https://itol.embl.de/).

### Meta-analyses of clinical manifestations

We systematically searched PubMed, Web of Science, and Google Scholar for published articles or reports without language restriction, which reported on the clinical manifestations in human infections in Europe from Jan 1, 2007, to Dec 31, 2023. The keywords used to search were "Chikungunya AND Europe". We retrieved published studies, including case reports, case series, and epidemiological reports for human infection. Two authors (Q. L. and H. S.) independently screened the titles and abstracts of all retrieved studies for inclusion, and then downloaded and assessed the full text of all eligible studies in the same manner. The final inclusion criteria focused on the clinical manifestations of chikungunya and their prevalence.

### Data analysis

A descriptive analysis was conducted to examine case characteristics, reporting countries, and countries of infection. Continuous variables were summarized as medians, and the categorical variables were estimated as the counts and proportion in each category. Poisson 95% Confidence Interval (95% CI) for the number counted was estimated using MedCalc software. All statistical analyses were carried out using R version 4.3.2 (https://cran.r-project.org/). Europe has traditionally been categorized into five regions, including Central, Eastern, Southern, Western and Northern Europe[21].

## Results

### Overall distribution from 2007 to 2023

The first recorded chikungunya case in mainland Europe was in 2007 in Italy. During 2007–2022, a total of 4730 chikungunya cases were reported in 22 countries in Europe

(Fig 1). The highest number of chikungunya cases was reported in the United Kingdom (1038, 21.9%), followed by France (929, 19.6%), Germany (684, 14.5%), and Itay (644, 13.6%) (S1 Table). There was a single peak in the overall yearly number of chikungunya cases. The most chikungunya cases were reported in 2014 (1461, 30.9%), followed by 2015 (624, 13.2%), 2017 (546, 11.5%), 2019 (535, 11.3%), and 2016 (488, 10.3%) (Fig 1). Several autochthonous outbreaks occurred, including in Italy in 2007 and 2014, France in 2014 and 2017, Spain in 2014, and the United Kingdom in 2014. No chikungunya cases were reported in Europe in 2023. The highest number of cases was reported in countries from Western Europe, followed by Southern Europe and Central Europe (S2 Table).

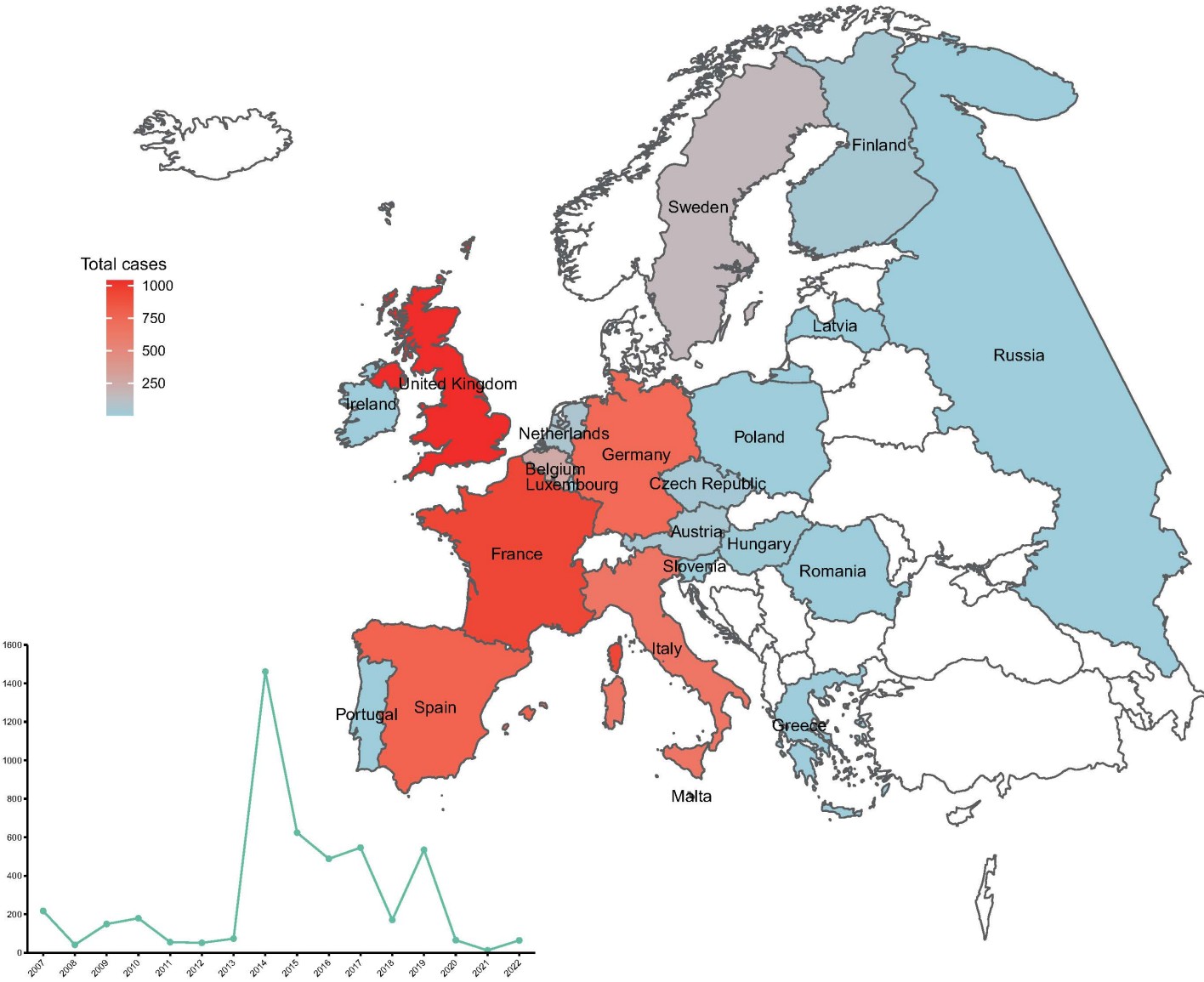

**Fig 1. Map illustrating the spatial distribution of chikungunya cases in mainland Europe.** The line plot in the lower left corner of the map shows the total number of chikungunya cases recorded each year. The map was constructed using R v4.3.2 software. The basemap shapefile of mainland Europe was downloaded from the GitHub website (https://github.com/leakyMirror/map-of-europe).

## Epidemiological characteristics

The majority of chikungunya cases from 2007 to 2022 were female (> 50%). Except for 2017 and 2021, the number of female chikungunya cases was higher than that of male cases in each year in Europe (Fig 2A). Six subgroups (0–4, 5–14, 15–24, 25–44, 45–64, and > 65) were divided according to age. The most cases were observed in the 25–44 and 45–64 age groups (Fig 2B). In 2008, 2011, 2014, 2017, 2020 and 2022, the largest number of chikungunya cases were in the 45–64 age group. The years with highest age-standardized incidence rate was 2014 (0.31 cases per 100 000 population), followed by 2015 (0.13), 2019 (0.10), and 2016 (0.10), with less than 0.1 in other years. The age-standardized incidence rate reported for each country per year was highest in Belgium in 2014 (68.6%), followed by Sweden in 2019 (57.8%), Spain in 2014 (56.7%), France in 2014 (55.0%), and Belgium in 2019 (53.0%) (Fig 2C and S3 Table).

## Travel-related chikungunya cases

In August 2007, an outbreak of chikungunya fever was reported in Italy with 217 laboratory-confirmed cases, indicating that the autochthonous transmission of CHIKV followed its introduction by a single returning visitor from India. From 2008 to 2011, the proportion of autochthonous cases decreased, but then reached its first peak in 2011 at 14.5% (S1 Fig). The largest proportion of autochthonous cases peaked in 2014 at 25.7%, accompanied by a high number of imported cases. Almost all chikungunya cases were travel-related resident who returned from non-European regions. It is important to note that the ranking of autochthonous case numbers did not always match the corresponding proportion.

To decipher the travel-related countries, a Sankey diagram was used to illustrate the major countries each year (Fig 3). Among the total travel-related cases (n=2836), most were acquired in North America (n=835), followed by Asia (n=672) and South America (n=316). The data

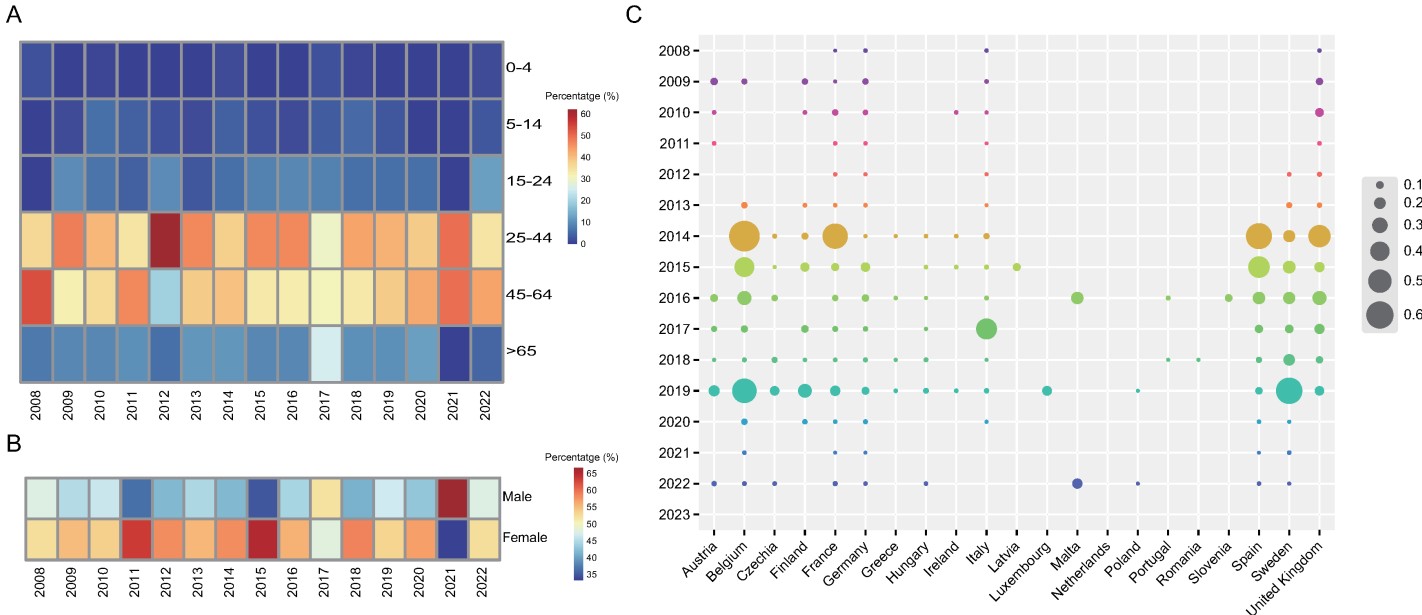

**Fig 2. Demographic characteristic and temporal distribution of chikungunya cases in Europe. (A)** The proportion of cases in the corresponding age subgroup. **(B)** The proportion of cases in the corresponding sex subgroup. **(C)** Bubble plot showing the temporal distribution of age-standardized incidence rates for each country per year. Data for the Netherlands is unavailable.

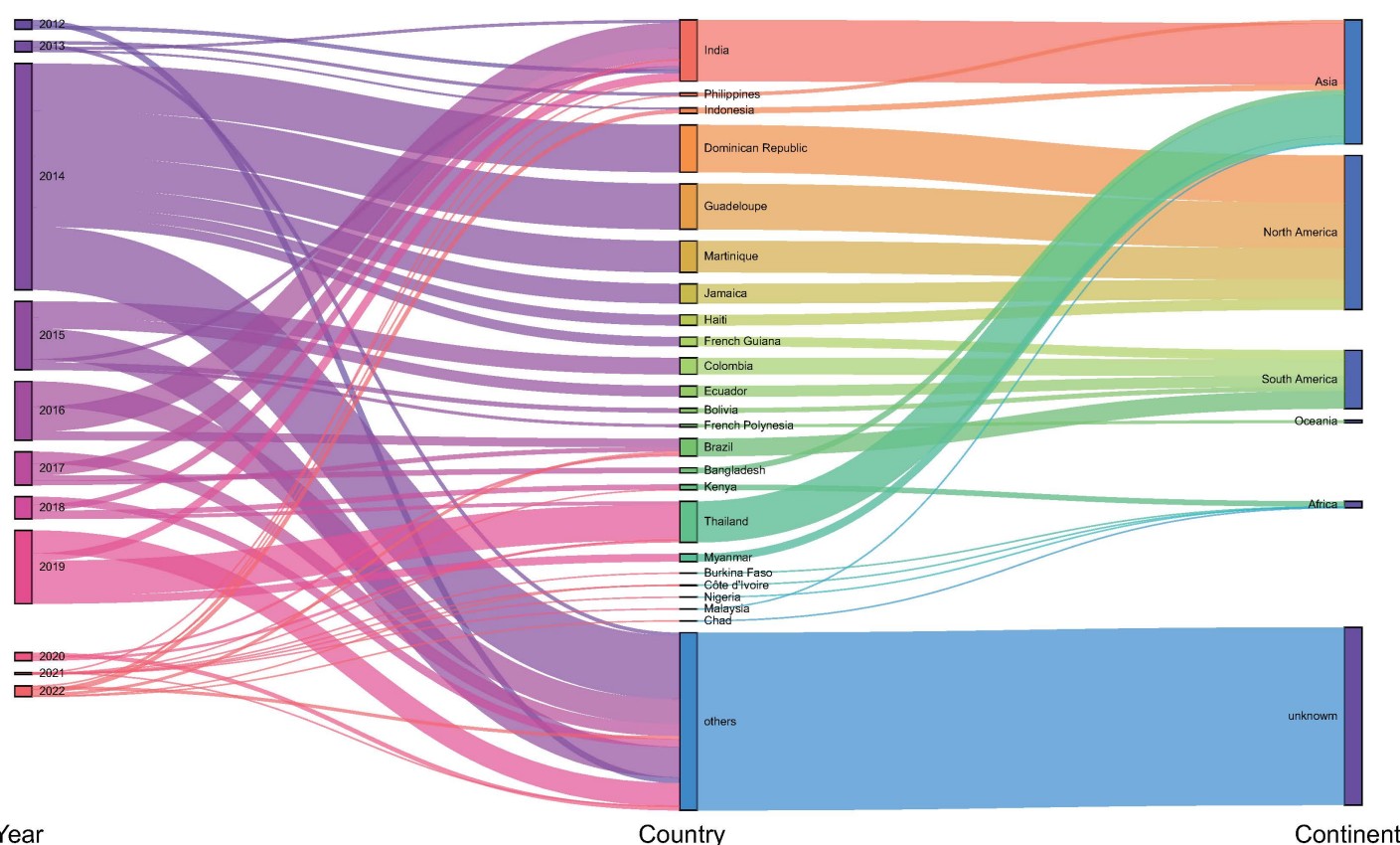

**Fig 3. Sankey diagram revealing the correspondence of travel-related chikungunya cases.** The diagram shows the collected year, probable country of infection, and continents in separate columns.

regarding the probable country of infection for 963 cases was unavailable. The majority of chikungunya cases reported in Europe had returned from 23 countries and territories. Specifically, the probable countries with the highest number of returned infections were India (332, 11.7%), followed by Dominican Republic (256, 9.0%), Guadeloupe (247, 8.7%), Thailand (222, 7.8%), and Martinique (170, 6.0%). It is worth mentioning that travel-related countries, including Dominican Republic, Guadeloupe, and Martinique, were only reported in 2014. Travelers that related to Idia were identified during 2015 to 2022. Travelers that related to Thailand were reported during 2018–2021, with the majority occurring in 2019.

## Seasonal variations

The total number of CHIKV disease cases varied each month, peaking in late summer between June and September. The lowest number of chikungunya cases was observed in March. The largest peak was seen in September (n=643), followed by June (n=529), July (n=484), and August (n=478) (S2 Fig). During the largest outbreak in 2017, the number of cases reported in September was considerably higher than the mean number of cases and even exceeded the maximum number of cases reported for the same period during 2008–2022 (S2 Fig). In the second largest outbreak of 2014, the number of cases reported for June-August exhibited peaks. However, the specific peak periods varied among reporting countries (S4 Table). May to August for France; September for Italy, May to July for Spain, September to December for the United Kingdom, October to January for Germany, July to September

for Belgium. Therefore, chikungunya cases in Europe showed a seasonal pattern with peaks mainly associated with summer vacation.

## Clinical manifestations

The systematic literature review was conducted on 1 August 2024, resulting in 645 records hits. After screening titles and abstracts, a total of 25 studies were found to be suitable for the meta-analysis of clinical manifestations. The pooled data formed an integrated database that included 800 human cases of CHIKV infections reported between 2009 and 2019 from 13 countries in Europe, primarily Italy (n=618), Spain (n=131), France (n=24), and Germany (n=9). Fever (proportion 97.6%; 95%CI 90.9–104.7), joint pain (94.3%; 95%CI 87.6–101.2), fatigue (63.5%; 95%CI 58.1–69.3), skin rash (52.3%; 95%CI47.4–57.5), muscle pain (47.4%; 95%CI 42.7–52.4), and headache (43.5%; 95%CI 39.1–48.3) were the most frequent (> 43%) among the 12 clinical symptoms recorded (Fig 4). Additionally, arthritis, photophobia, conjunctivitis, diarrhea, itching, and vomiting were also observed but in a small number of chikungunya cases.

## Genomic surveillance of CHIKV

We utilized 89 complete or partial CHIKV genome sequences with known hosts in our analysis, including 84 from human patients and 5 from mosquitoes. Data was contributed by eight countries, with the majority from Italy (n=64, 71.9%), followed by France (n=14), Finland (n=4), Ireland (n=2), Germany (n=2), Russia (n=1), Spain (n=1), and the United

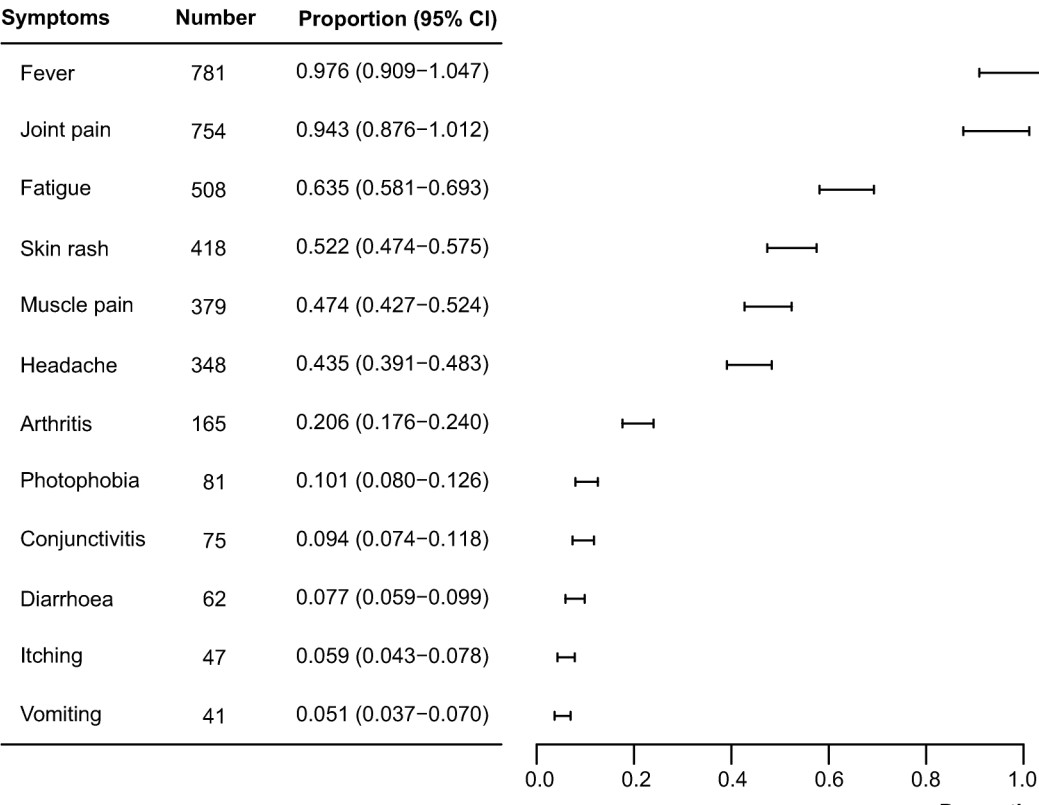

| Symptoms | Number | Proportion (95% CI) |
|---|---|---|
| Fever | 781 | 0.976 (0.909−1.047) |
| Joint pain | 754 | 0.943 (0.876−1.012) |
| Fatigue | 508 | 0.635 (0.581−0.693) |
| Skin rash | 418 | 0.522 (0.474−0.575) |
| Muscle pain | 379 | 0.474 (0.427−0.524) |
| Headache | 348 | 0.435 (0.391−0.483) |
| Arthritis | 165 | 0.206 (0.176−0.240) |
| Photophobia | 81 | 0.101 (0.080−0.126) |
| Conjunctivitis | 75 | 0.094 (0.074−0.118) |
| Diarrhoea | 62 | 0.077 (0.059−0.099) |
| Itching | 47 | 0.059 (0.043−0.078) |
| Vomiting | 41 | 0.051 (0.037−0.070) |

**Fig 4. Meta-analysis of the different clinical manifestations of chikungunya cases.**

Kingdom (n=1). Most of the available CHIKV genomes were collected between 2006 and 2019, with a significant portion from 2017 (n=43), 2006 (n=13), and 2007 (n=13) during the initial and recent epidemics. Genotyping results revealed that the majority of these genomes belonged to II-ECSA (78, 87.8%), with remainder being by III-Asian (11, 12.2%). The five CHIKV genomes taken from mosquitoes were classified as II-ECSA. Phylogenetic analysis also indicated that the circulating CHIKV could be clustered into two clades: II-ECSA and III-Asian (Fig 5). However, specific years with high reported cases, such as 2016 and 2017, lacked genomic data genomic data (Fig 5B). Furthermore, there were unavailable genomic data in most countries where chikungunya cases were reported (Figs 5B and S3).

## Discussion

The pandemic potential of CHIKV, an *Aedes* mosquito-borne alphavirus, has long been recognized. The effects of climate change, as well as increased globalization of commerce and travel, have expanded the habitat of *Aedes* mosquitoes, leading to an increased number of people at risk of contracting chikungunya fever in the coming years [22]. Local outbreaks of chikungunya in temperate areas of Europe have occurred in the last two decades [23,24]. Here, we conducted an extensive investigation into CHIKV, including spatial distribution, epidemiology, clinical manifestations, and phylogenetics.

Before the first autochthonous cases of CHIKV was reported in mainland Europe in 2007 [25], a serious outbreak of chikungunya fever occurred between March 2005 and June 2006 in the French overseas territories [26]. However, the study focuses on the population affected by chikungunya fever in mainland Europe, where 4730 human infections were reported between 2007 to 2022. Although no chikungunya case reported in 2023, non-travel associated chikungunya cases were reported in France in 2024, emphasizing that the chikungunya outbreak is still ongoing. Our descriptive characteristics provided insight into establishing a comprehensive history of CHIKV in Europe, especially regarding the variability of chikungunya cases

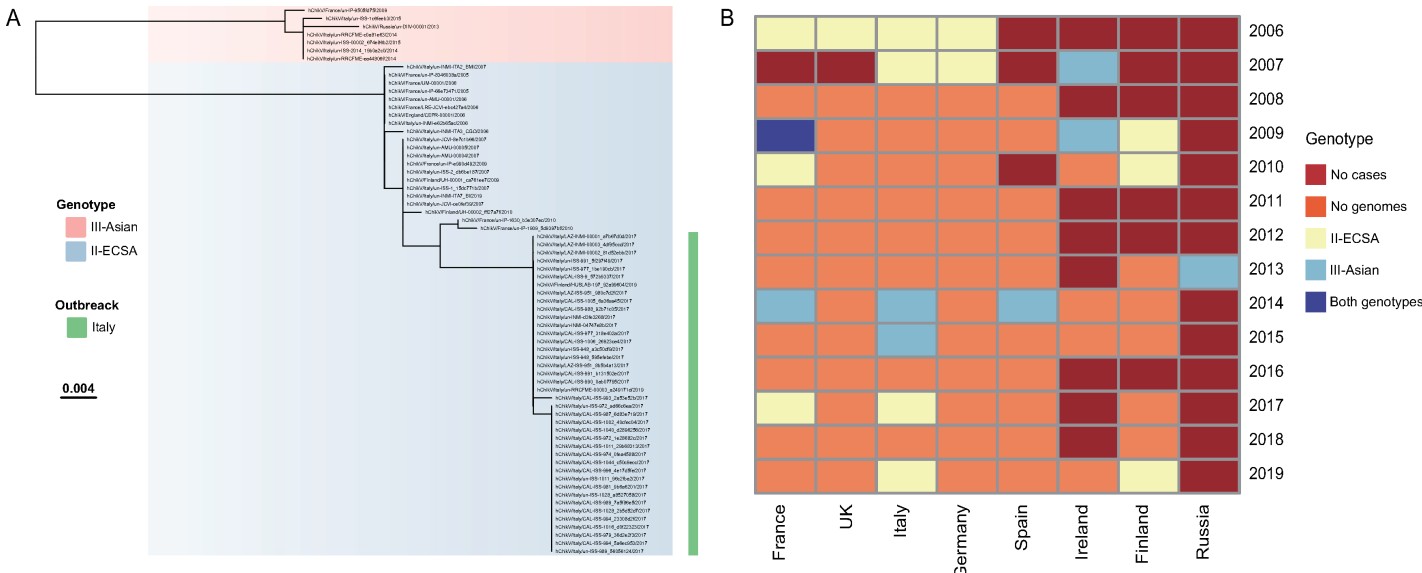

**Fig 5. Phylogenetic and genotyping analyses. (A)** A maximum-likelihood phylogenetic tree based on the E1 glycoprotein. After aligned, a total of 69 publicly available chikungunya virus genomes are used in the analysis. The chikungunya virus sequences collected during the 2017 outbreak in Italy are labeled with a blue column. **(B)** Spatiotemporal distribution of chikungunya virus genotypes. All available genome sequences of chikungunya virus are used.

across different regions and over time. The transmission dynamics of CHIKV showed that the two largest outbreaks occurred in 2017 and 2014, with France and Italy being the most affected countries, respectively. Almost half of the countries in Europe have reported chikungunya cases, while the remaining countries are equally at risk.

Although the number of chikungunya cases has fluctuated over the years, the majority of cases in Europe were highly travel-related each year except for 2007. The highest number of cases was reported in 2014, with 74% of them being travel-related. Most cases in 2014 were individuals returning from the Americas, where a large outbreak of CHIKV disease occurred during that time [27]. In the remaining years, the vast majority of chikungunya cases were travel-related, with the proportion being higher than 95% after 2014. During the COVID-19 pandemic (2020–2023), the number of chikungunya cases was significantly lower that before the pandemic (2017–2019). This was most likely due to non-pharmaceutical interventions such as travel restrictions and border closure introduced during the COVID-19 outbreak [28], which reduced the number of Europeans travelling across borders.

Both climate and vector factors should be considered for CHIKV. In 2007, the first identified autochthonous outbreak of chikungunya fever in Europe was imported to France by a single returning visitor to India, demonstrating the potential of the *Aedes albopictus* mosquito to transmit the virus in a temperate European climate [29]. Alongside *Ae. aegypti*, *Ae. albopictus* is one of the most dangerous mosquito vectors of arbovirus, acting as a primary vector of chikungunya. *Ae. albopictus* was transported from Italy to France in 2000, and then became endemic [30]. Over the following 10 years, this vector expanded to Eastern Spain, Switzerland, Monaco, Malta, Balkan countries, and Greece [30]. Additionally, the economies of European countries are more developed, and the per capita income level is higher, contributing to the globalization of travel and trade.

Fever and joint pain (arthralgia) were the most prevalent symptoms, affecting 97.6% and 94.3% of symptomatic cases, respectively, which were higher than the recently reported global prevalence [12]. The third most common symptom, fatigue, was found to be different from that of global prevalence [12] and previous reports (skin rash) [22], which indicated skin rash as a prominent symptom. Furthermore, additional symptoms can also occur in CHIKV infection, as described in previous literature as common for chikungunya [12]. Despite clear inclusion criteria being used, these estimates were likely affected by selection bias. The estimates of chikungunya cases reported were mainly based on patients with healthcare-seeking behavior, thus excluding asymptomatic patients. The co-circulation of several different arboviral diseases with similar clinical manifestations (dengue virus and zika virus) posed a challenge for clinicians to accurately diagnose without laboratory testing [3], potentially leading to an underestimation of reported chikungunya cases. Therefore, estimates in Europe were likely to be underestimated. Additionally, chikungunya infection may result in severe symptoms, particularly in female aged 45–64 years. However, deaths caused by CHIKV were rarely reported in Europe with only one death occurring between 2007 and 2012 [25].

The chikungunya genomes collected from Europe were classified into two genotypes, with the majority being II-ECSA and the absence of the West African genotype. The first instance of A226V mutation reported in Europe was detected in Italy in 2007, and later identified in France in 2014 and 2017. The majority of genomes were sequenced from Italy in 2017, which lacked the E1 A226V mutation [31]. This 2017 strain was introduced and rapidly spread in Southern Italy, fully adapting to the *Ae. albopictus* mosquito vector compared with the 2007 mutated strain [32]. Due to travel-related cases and biased sequencing, the time of most recent common ancestor of CHIKV in Europe was not estimated in this study. Furthermore, the spatiotemporal transmission patterns in Europe or local transmission within nations were also not conducted due to the lack of insufficient complete genomes.

## Conclusions

In conclusion, this study presents a comprehensive epidemiological characteristics and potential risks of chikungunya virus in Europe. Given the ongoing chikungunya outbreaks worldwide, the most effective tools to combat the disease in Europe involve surveillance and vector control.

## Supporting information

**S1 Fig.** Line plot showing the proportion of travel-related chikungunya cases recorded each year in Europe.
(TIF)

**S2 Fig.** Number of chikungunya cases per month during 2008–2023 in Europe.
(TIF)

**S3 Fig. Bubble plot revealing the temporal distribution of number of chikungunya cases for each country per year.**
(TIF)

**S1 Table. Number of chikungunya cases reported for each country per year between 2007 and 2023.**
(DOCX)

**S2 Table. Comparison of the number of chikungunya cases by regions of Europe.**
(DOCX)

**S3 Table. Age-standardized rate of chikungunya cases reported for each country per year between 2008 and 2023.**
(DOCX)

**S4 Table. Number of cases per country per month in 2014.**
(DOCX)

## Acknowledgments

We thank all the originating laboratories that generated chikungunya virus sequences, and also acknowledge the GISAID database for providing the open-sharing platform.

## Author contributions

**Conceptualization:** Qian Liu, Li Gu, Hui Yuan, Wentao Zhu.

**Data curation:** Qian Liu, Wentao Zhu.

**Formal analysis:** Qian Liu, Hong Shen.

**Funding acquisition:** Wentao Zhu.

**Investigation:** Qian Liu, Hong Shen, Wentao Zhu.

**Methodology:** Qian Liu, Wentao Zhu.

**Project administration:** Wentao Zhu.

**Resources:** Qian Liu, Wentao Zhu.

**Software:** Qian Liu.

**Supervision:** Li Gu, Hui Yuan, Wentao Zhu.

**Validation:** Wentao Zhu.

**Visualization:** Qian Liu.

**Writing – original draft:** Qian Liu, Wentao Zhu.

**Writing – review & editing:** Qian Liu, Hong Shen, Li Gu, Hui Yuan, Wentao Zhu.

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
