## [Decision Letter · Decision Letter 0]

22 Jan 2025

PNTD-D-24-01472Chikungunya virus in Europe: a retrospective epidemiology study from 2007 to 2023PLOS Neglected Tropical Diseases Dear Dr. Zhu, Thank you for submitting your manuscript to PLOS Neglected Tropical Diseases. After careful consideration, we feel that it has merit but does not fully meet PLOS Neglected Tropical Diseases's publication criteria as it currently stands. Therefore, we invite you to submit a revised version of the manuscript that addresses the points raised during the review process. Please submit your revised manuscript within 30 days Mar 23 2025 11:59PM. If you will need more time than this to complete your revisions, please reply to this message or contact the journal office at plosntds@plos.org. Please include the following items when submitting your revised manuscript: * A rebuttal letter that responds to each point raised by the editor and reviewer(s). You should upload this letter as a separate file labeled 'Response to Reviewers '. This file does not need to include responses to any formatting updates and technical items listed in the 'Journal Requirements' section below. * A marked-up copy of your manuscript that highlights changes made to the original version. You should upload this as a separate file labeled 'Revised Manuscript with Track Changes '. * An unmarked version of your revised paper without tracked changes. You should upload this as a separate file labeled 'Manuscript '. If you would like to make changes to your financial disclosure, competing interests statement, or data availability statement, please make these updates within the submission form at the time of resubmission. Guidelines for resubmitting your figure files are available below the reviewer comments at the end of this letter. We look forward to receiving your revised manuscript. Kind regards, Eric Mossel, Ph.D.Academic EditorPLOS Neglected Tropical Diseases Michael HolbrookSection EditorPLOS Neglected Tropical Diseases

Shaden Kamhawi

co-Editor-in-Chief

Paul Brindley

co-Editor-in-Chief

**Journal Requirements:**

At this stage, the following Authors/Authors require contributions: Qian Liu, Hong Shen, Li Gu, and Wentao Zhu. Please ensure that the full contributions of each author are acknowledged in the "Add/Edit/Remove Authors" section of our submission form.

Potential Copyright Issues:

- Figure 1. Please (a) provide a direct link to the base layer of the map (i.e., the country or region border shape) and ensure this is also included in the figure legend; and (b) provide a link to the terms of use / license information for the base layer image or shapefile. We cannot publish proprietary or copyrighted maps (e.g. Google Maps, Mapquest) and the terms of use for your map base layer must be compatible with our CC BY 4.0 license.

5) We note that your Data Availability Statement is currently as follows: "Not applicable.". Please confirm at this time whether or not your submission contains all raw data required to replicate the results of your study. Authors must share the “minimal data set” for their submission. PLOS defines the minimal data set to consist of the data required to replicate all study findings reported in the article, as well as related metadata and methods (https://journals.plos.org/plosone/s/data-availability#loc-minimal-data-set-definition).

- The points extracted from images for analysis..

6) Please ensure that the funders and grant numbers match between the Financial Disclosure field and the Funding Information tab in your submission form. Note that the funders must be provided in the same order in both places as well.

**Reviewers' comments:** Reviewer's Responses to Questions

**Key Review Criteria Required for Acceptance?**

**Methods**

-Are the objectives of the study clearly articulated with a clear testable hypothesis stated?

-Is the study design appropriate to address the stated objectives?

-Is the population clearly described and appropriate for the hypothesis being tested?

-Is the sample size sufficient to ensure adequate power to address the hypothesis being tested?

-Were correct statistical analysis used to support conclusions?

-Are there concerns about ethical or regulatory requirements being met?

Reviewer #1: The review article examined existing available data and used applied basic statistical analyses. I have no concerns.

Reviewer #2: -Are the objectives of the study clearly articulated with a clear testable hypothesis stated?

Yes, although they should have included deepening climate change

-Is the study design appropriate to address the stated objectives?

Yes, since there is information in the publications of epidemiological surveillance in Europe.

-Is the population clearly described and appropriate for the hypothesis being tested?

Yes

-Is the sample size sufficient to ensure adequate power to address the hypothesis being tested?

Yes

-Were correct statistical analysis used to support conclusions?

Yes

-Are there concerns about ethical or regulatory requirements being met?

People's rights are not being violated, the information that is available is not nominalized.

**Results**

-Does the analysis presented match the analysis plan?

-Are the results clearly and completely presented?

-Are the figures (Tables, Images) of sufficient quality for clarity?

Reviewer #1: The results of the analyses provide an epidemiological review of the target organism within the defined timeline. The data are presented well and with clarity. I have no concerns.

Reviewer #2: The analysis presented coincides with the plan.

Line 203 to 206.I think this information would look better in a table distributing the countries and months of specific peaks.

**Conclusions**

-Are the conclusions supported by the data presented?

-Are the limitations of analysis clearly described?

-Do the authors discuss how these data can be helpful to advance our understanding of the topic under study?

-Is public health relevance addressed?

Reviewer #1: The conclusions are reasonable and defended with the data. There are no areas of over-statement and the conclusions seem void of bias. I have no concerns.

Reviewer #2: An integrated strategy is essential.

Undoubtedly, the increase in travel and the transit of goods and people can be a risk factor to take into account, but it is not the only one. Climate change is positioning itself as a more important base factor. For this reason, I do not think it is appropriate to justify the reduction of travel to prevent the transmission of CHIKV.

Line 282 to 283. This statement should be supported by entomological data on the infestation of both vectors over the years monitored. If there is no information on the matter, it could not be affirmed, the infestation by both vectors collaborates with the increase in transmission.

**Editorial and Data Presentation Modifications?**

Reviewer #1: 36 Guadeloupe8.7%) = please add space between

56 Guadeloupe8.7%), = please add space between

141 Europe has traditionally been categorized into 142 five regions, including Central, Eastern, Southern, Western and Northern Europe. = This could use a reference to define these boundaries. Perhaps a bit petty, but I think it helps to ensure the epidemiological analysis is clearly defined by solid reference for these exact boundaries.

Reviewer #2: I believe that the definitions expressed here should be clarified and separated. There should be a definition of probable case and one for confirmed case. Also for any of them add the date of return to the country.

LIne 187 to 190. These data are not clear from what was said in the previous sentence, where it is said that most of them were acquired in North America.

**Summary and General Comments**

Reviewer #1: The presented work was a pleasure to read. I would like to thank the author’s for submitting a very polished and complete manuscript for review. The work provides valuable insight following the careful review and statistical and comparative analyses of available data from 2007-2023.

Reviewer #2: (No Response)

PLOS authors have the option to publish the peer review history of their article (what does this mean? ). If published, this will include your full peer review and any attached files.

**Do you want your identity to be public for this peer review?** For information about this choice, including consent withdrawal, please see our Privacy Policy .

Reviewer #1: **Yes: ** Aaron Timothy Phillips

Reviewer #2: No

---

## [Editor Report · Decision Letter 1]

7 Feb 2025

Dear Dr. Zhu,

We are pleased to inform you that your manuscript 'Chikungunya virus in Europe: a retrospective epidemiology study from 2007 to 2023' has been provisionally accepted for publication in PLOS Neglected Tropical Diseases.

Best regards,

Michael R Holbrook, PhD

Section Editor

Michael Holbrook

Section Editor

Shaden Kamhawi

co-Editor-in-Chief

Paul Brindley

co-Editor-in-Chief

---

## [Editor Report · Acceptance letter]

Dear Dr. Zhu,

We are delighted to inform you that your manuscript, "Chikungunya virus in Europe: a retrospective epidemiology study from 2007 to 2023," has been formally accepted for publication in PLOS Neglected Tropical Diseases.

Best regards,

Shaden Kamhawi

co-Editor-in-Chief

Paul Brindley

co-Editor-in-Chief
